# Assessment of Treated Wastewater Reuse in Drip Irrigation under Different Pressure Conditions

**Naji K. Al-Mefleh** [1,*], **Samer Talozi** [2] and **Khalid Abu Nasir** [3]

1. Department of Natural Resources, Faculty of Agriculture, Jordan University of Science and Technology, Irbid P.O. Box 3030, Jordan
2. Civil Engineering Department, Faculty of Engineering, Jordan University of Science and Technology, Irbid P.O. Box 3030, Jordan; samerbse@just.edu.jo
3. Department of Natural Resources and Management, Faculty of Agriculture, Irbid P.O. Box 3030, Jordan; Khaledmohammad2010@hotmail.com
* Correspondence: nmefleh@just.edu.jo

**Abstract:** This study aims to investigate the influence of treated wastewater (TWW) on the hydraulic performance of drip irrigation emitters. A field experiment was conducted in order to test two types of online emitters, a low pressure (LP) and a standard pressure (SP), at different working pressures (0.25 bar, 0.50 bar, and 1.00 bar) using TWW. The emitters were initially evaluated in the laboratory and the field for the discharge exponent ($X$), discharge coefficient ($Kd$), average emitter discharge ($Q_{avg}$), coefficient of variation ($CV$), distribution uniformity ($DU$), the mean discharge ratio ($Dra$), and the main degree of clogging ($DC$). The main effect of the emitters on the hydraulic parameters of irrigation performance was not significant, while the operational pressure and operational time of irrigation had a significant effect. For the LP emitter, the average emitter discharge was 7.6, 7.7, and 7.8 Lh$^{-1}$ at 0.25, 0.50, and 1.00 bar, respectively. For the SP emitter, the average emitter discharge was 7.6, 7.8, and 7.8 Lh$^{-1}$ at 0.25, 0.50, and 1.00 bar, respectively. The *EU* values for the LP and SP emitters varied from low to moderate at 0.25 bar, as the EU values at 0.50 and 1.00 bar were considered high for both emitter types.

**Keywords:** drip irrigation; treated wastewater; clogging emitters; compensated emitters; emitter characteristics

## 1. Introduction

The issue of water scarcity has placed stress on available water resources in Jordan. Due to the arid and semi-arid climate, the adverse effects of climate change, and a growing population, there has been an increasing demand for the available freshwater resources, affecting water quantity and quality [1]. Jordan relies on groundwater recharge from rainfall for water supplies, although around 94% of the precipitation evaporates [2]. Moreover, Jordan's agricultural sector has used 65% to 70% of Jordan's freshwater resources [3]. These obstacles urge the water authorities to seek alternatives in order to save the country's limited water supply.

By using treated wastewater (TWW) and high-efficiency irrigation methods, such as drip irrigation, the agricultural sector has the opportunity to decrease its significant freshwater intake, helping to conserve the resource and mitigate the effects of increasing water demand. Treating wastewater and reusing it for irrigation further reduces the demand for freshwater resources [4]. Moreover, the availability of nutrients in wastewater, such as phosphorus, nitrogen, and potassium, has the potential to increase crop yield and reduce the use of synthetic fertilizers, which is an added incentive for farmers to use TWW as a source for irrigation [5]. These advantages are maximized if the clogging potential for emitters is addressed and avoided. Emitter clogging is one of the most critical factors that affect the performance of drip irrigation systems. The causes of clogging are categorized

into physical, biological, and chemical. Physical deposits when using TWW are mainly composed of organic matter, clay, and silt-sized aggregates [6].

Drip irrigation provides a solution to the water crisis all over the world. It has numerous advantages such as the promotion of plant growth, increased yields, reduction of soil salinity in the plant root area, reduction of weed growth, and suitability for desert climates with improved water use efficiencies. According to a World Bank (2006) report, drip irrigation uses 30–50% less water than surface irrigation, reduces salinization and waterlogging, and achieves up to 95% irrigation efficiency rates [7]. In Jordan, drip irrigation systems cover an area of 92,500 hectares of the total area suitable for agriculture, with an annual increase of 0.1% [8]. However, drip irrigation efficiencies are restricted if there is persistent clogging of the emitters, especially under the use of TWW.

Chemical precipitation may occur with the use of TWW for irrigation depending on water quality parameters such as temperature, pH, and salt concentrations. These parameters could induce the precipitation and sedimentation of mineral elements that generate clogging in drip emitters. Biological clogging is often coupled with physical and chemical deposits. The flushing of drip lines can clear the buildup of inorganic and organic materials in emitter flow paths and on the inside walls of the drip emitters [9]. Nakayama et al. found that flushing must occur frequently and at an appropriate velocity in order to dislodge and transport accumulated sediments and effectively slow down the clogging of emitters [10]. Nevertheless, clogging is a common problem while using TWW for drip irrigation. It has a direct influence on the performance and service life of a drip irrigation system and significantly increases the non-uniformity of water distribution in the irrigated field, which has a negative impact on the growth and overall yield of crops [11,12].

Pressure is a significant factor affecting the performance of a drip irrigation system. Studies have shown that the largest pressure loss in the drip irrigation system occurs in the pressure compensating emitters [13]. Additionally, most commercial compensating emitters have a minimum operating pressure between 0.50 bar and 1.00 bar [14]. In order to address the effect of pressure on drip irrigation efficiency, two online pressure-compensating emitters have been tested under different pressure conditions. The first is a low-pressure emitter (LP) emitter that delivers $8 \, L \, h^{-1}$ at an activation pressure of 0.15 bar, and the second is a standard pressure (SP) emitter that delivers $8 \, L \, h^{-1}$ at 1.00 bar. Based on these characteristics, it is expected that using the low-pressure emitter will reduce pumping requirements, leading to energy savings while running the drip irrigation system. The main objective of this study is to evaluate the performance of these emitters in the field at different operational pressures (0.25, 0.50, and 1.00 bar) and over different operational times (with 20-h time intervals) using TWW. Under these conditions, different emitter performance characteristics are investigated, namely, average emitter discharge, emission uniformity (*EU*), coefficient of variation (*CV*), Christiansen uniformity (*CU*), and the mean degree of clogging (*DC*).

## 2. Materials and Methods

### 2.1. Preparation of the Experiment

The study was carried out near Al-Ramtha Agricultural Research Station, adjacent to Al-Ramtha Wastewater Treatment Plant, located 7 km north of Al-Ramtha (32° 35′ north latitude and 35° 59′ east longitude) and at an elevation of 490 m above sea level. The experiment took place from the beginning of March 2018 to the end of November 2018. The inlet of the Al-Ramtha Wastewater Treatment Plant was 100% TWW with a secondary mechanical treatment process, which employed the activated sludge-extended aeration method of treatment.

The experiment layout consisted of three blocks, each block containing three pressure sets. Each pressure set carried two laterals—one for LP emitters and the second for SP emitters. Emitters were installed 50 cm apart on a 20 m long lateral (40 emitters for each lateral). A pump was installed in order to provide the necessary pressure for the emission of water. The irrigation system consisted of one tank (180 $m^3$), a pump, valves, two

filters, pressure gauges, three flow meters, three reducer valves, and lateral pipes (20 mm diameter), as shown in the schematic diagram of the experiment in Figure 1.

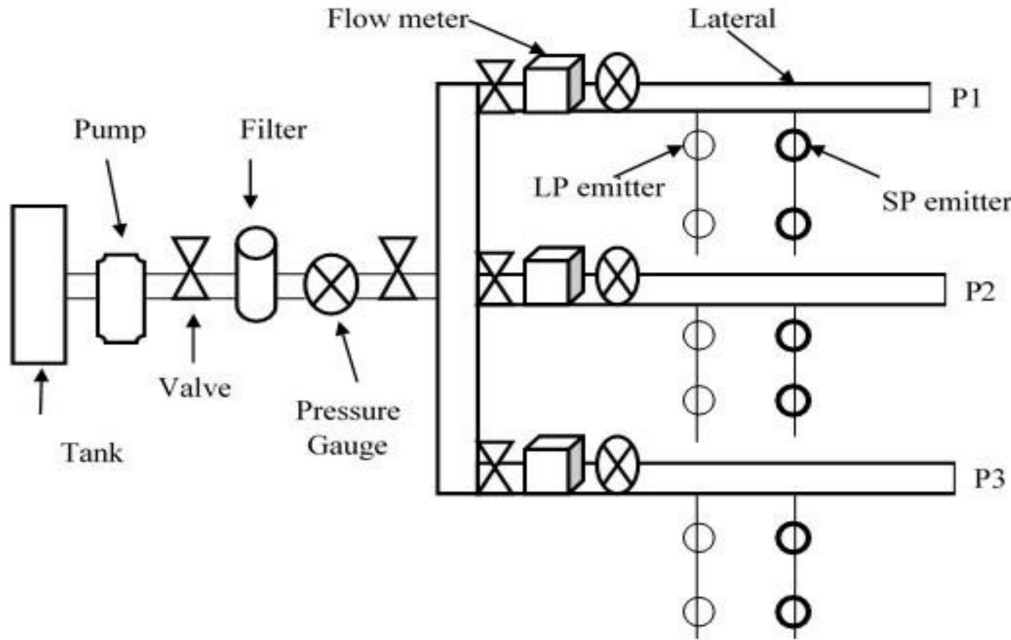

**Figure 1.** Schematic diagram of the irrigation system for testing the low-pressure (LP) emitter and standard pressure (SP) emitters under different operating pressures (P1= 0.25 bar, P2 = 0.50 bar, P3 = 1.00 bars) using treated wastewater (TWW).

### 2.2. Irrigation System

The two types of emitters that were tested using TWW were the LP emitter and the SP emitter. The specifications of the LP emitter are described as pressure compensating, delivering 8 L h$^{-1}$ at 0.15 bar. The SP emitter specifications are also described as pressure compensating but deliver 8 L h$^{-1}$ at 1.00 bar. Since the LP emitters did not give 8 L h$^{-1}$ under the TWW, it was decided to test the emitters at 0.25 bar instead of 0.15 bar. The TWW passed through the sand filter, followed by screen filter and disk filters, respectively. The sand filter consisted of a layer of gravel with a diameter of 8 to 16 mm and another gravel layer with a diameter of 1 to 8 mm. The screen filter consisted of a diameter of 1.6 mm and the disk filter had 250 mesh/in$^2$. The system flushing was based on the loss of pressure between the inlet and outlet for each filter and occurred before each irrigation process.

### 2.3. Water Resources and Qualities

The main parameters of TWW qualities are pH, electrical conductivity (EC), TDS, Ca, Mg, Na, K, Cl, Mn, SO$_4$, Fe, NO$_3$, P, B, biochemical oxygen demand (BOD$_5$), and chemical oxygen demand (COD). Water samples (1000 mL) were collected and tested in the laboratories of Methods of Irrigation and Agriculture (MIRRA) Association, at 7-Abdelaziz Al-Tha'alibi Str., Shmeisani, Amman 11194, Jordan. Electrical conductivity (EC) and pH meters were used to test EC and water reaction (pH) for each water sample. Other parameters (Ca, Mg, Na, K, Cl, Mn, SO$_4$, Fe, NO$_3$, P, B) were measured by flame spectrometry. Incubation techniques with oxygen determinations by the Winkler Method were used in order to test BOD$_5$ [15]. Micro digestion and colorimetric methods were used for the COD analysis. The TWW was passed through the sand filter followed by disk filters. The irrigation system was flushed for five minutes after each discharge measurement of the emitters.

### 2.4. Field Measurements

Emitter flow-rate tests were conducted at three different pressure intervals (P1 = 0.25, P2 = 0.50, and P3 = 1.00 bars). From each lateral, 20 out of 40 emitters were tested. Every other emitter on the lateral was selected. The emitter discharge was collected every ten minutes using a graduated cylinder and was converted to liters per hour. Every week, the system ran for two hours, three days per week. After an operational time of 20 h, a set of emitter discharge readings were measured. In total, eight sets of readings were recorded for estimating the parameters of emitter characteristics. Laterals were flushed with Phosphoric acid (ratio of 100 L water:2 L phosphoric acid 85%) every 20 h for five minutes at a time after the tests were conducted. The pressure was measured at the beginning and end of the lateral line by a pressure gauge. The lateral lines were raised using rigid iron rods at a height of 20 cm above the soil surface. Under each emitter, water pots with a capacity of three liters were used to collect the discharge of emitters.

At the beginning of the experiment, the emitter characteristics of the initial values of average discharge ($Q_{avg}$), discharge coefficient ($Kd_{in}$), and discharge exponent ($X_{in}$) were estimated for the new emitters. At the end of the experiment, the final ($f_i$) average discharge $Q_{avg}$, $Kd_{fi}$, and $X_{fi}$ were estimated. The values of $X$ and $Kd$ were derived from the equation of flow as a function of pressure as follows:

$$Q = Kd * (P)^X. \tag{1}$$

The averages of $Q_{avg}$, *CV*, *EU*, *CU*, and *DC* were estimated after each operational time interval at each pressure setting. The overall averages of these parameters were estimated as averages of eight tests (160 h of operational time with 20-h intervals).

### 2.5. Estimating Hydraulic Parameters of Emitter Performance Characteristics

The main emitter flow characteristics are estimated according to the equations presented in Table 1.

**Table 1.** Equations for estimating the hydraulic parameters.

| Hydraulic Parameters | Equations | |
|---|---|---|
| Average emitter discharge ($Q_{avg}$) | $Q_{avg} = \dfrac{\sum\limits_{i=1}^{n} q_i}{n}$ | (2) |
| Coefficient of variation (*CV*) [16] | $CV = \dfrac{SD}{Q_{avg}} * 100$ | (3) |
| Emission uniformity (*EU*) [17] | $EU = \dfrac{q_{avg\frac{1}{4}}}{q_{avg}} * 100$ | (4) |
| Christiansen uniformity (*CU*) [18] | $CU = \left(1 - \dfrac{\sum\limits_{i=1}^{n} \left|q_i - q_{avg}\right|}{n * q_{avg}}\right)$ | (5) |
| The main degree of clogging (*DC*) | $DC = \left(1 - \dfrac{\sum\limits_{i=1}^{n} q_i}{n\bar{q}_{new}}\right) * 100$ | (6) |
| The discharge exponent (*X*), [19] | $X = \dfrac{\log\left(\dfrac{Q_{avg1}}{Q_{avg2}}\right)}{\log\left(\dfrac{H_{avg1}}{H_{avg2}}\right)}$ | (7) |
| The discharge coefficient (*Kd*), [19] | $Kd = \left(\dfrac{Q_{avg}}{H^x}\right)$ | (8) |

Where $\sum q_i$ is the sum of the individual discharge in each lateral line; $n$ is the number of emitters; *SD* is the standard deviation of emitter discharge; $Q_{avg}$ is the mean discharge of the emitter in the same lateral (Lh$^{-1}$);$q_{avg\frac{1}{4}}$ is the average rate of discharge of the lowest quarter of the field data of emitter discharge; $q_i$ is the individual emitter discharge (Lh$^{-1}$); $\bar{q}_{new}$ is the average discharge when the emitter was new (Lh$^{-1}$); $Q_{avg1}$ is the average discharge at operational pressure $H_{avg1}$(0.25 bar); $Q_{avg2}$ is the average discharge at operation pressure $H_{avg2}$ (1.00 bar).

## 3. Results and Discussion

### 3.1. Water Quality

Five samples of TWW were taken on a monthly basis from March 2018 to November 2018. The mean values for each chemical and biological water quality parameter for TWW are presented in Table 2. The clogging risk was evaluated according to the classification proposed by the water quality criteria for emitter clogging [12,20,21]. The water quality parameters of pH (8.2), TDS, Mn, and Fe were used as a guide to determine the potential of emitter clogging. The pH values show a severe potential to cause emitter clogging for any water type. This is consistent with the findings of Al-Mefleh et al., and Al-Mefleh and Al-Raja [21,22]. However, the hardness characteristic (Ca and Mg) is another factor that might cause the precipitation of carbonate, leading to an increase in the potential of emitter clogging. The hardness value of the TWW was estimated to be about 428 mg/L, which is defined as hard water (>200 mg/L). Increasing the pH and temperature can increase the precipitation of Ca and Mg in drip irrigation systems.

**Table 2.** Chemical and biological analysis of the TWW.

| Parameter | Units | Means [a] | Clogging Potential Limits [b] [20] | | |
|---|---|---|---|---|---|
| | | | L* | M* | H* |
| pH | | 8.2 ($\pm$0.28) | <7.0 | 7.0–8.0 | >8.0 |
| EC [c] | dS/m | 2.6 ($\pm$0.08) | - | - | - |
| TDS [d] | mg/L | 1707($\pm$51) | <500 | 500–2000 | >2000 |
| Ca | mg/L | 98.6 ($\pm$42.2) | - | - | - |
| Mg | mg/L | 44.3 ($\pm$1.94) | - | - | - |
| Na | mg/L | 296 ($\pm$19.3) | - | - | - |
| K | mg/L | 44.3 ($\pm$2.61) | - | - | - |
| Cl | mg/L | 211.4 ($\pm$26.2) | - | - | - |
| Mn | mg/L | 0.47 ($\pm$0.039) | <0.1 | 0.1–1.5 | >1.5 |
| $SO_4$ | mg/L | 81.4 ($\pm$13) | - | - | - |
| Fe | mg/L | 0.11 ($\pm$0.04) | <0.2 | 0.2–1.5 | >1.5 |
| P | mg/L | 3.6 ($\pm$0.72) | - | - | - |
| $NO_3$ | mg/L | 0.51 ($\pm$0.23) | - | - | - |
| B | mg/L | 0.3 ($\pm$0.08) | - | - | - |
| COD | mg/L | 58.6 ($\pm$26.3) | - | - | - |
| $BOD_5$ | mg/L | 16 ($\pm$8.89) | - | - | - |

[a] Standard deviation at confidence level of 95%. [b] Clogging potential [12,20,21]. [c] Electrical conductivity. [d] Total dissolved solids. L*: low. M*: medium. H*: high.

The mean values of EC varied from 2.5 to 2.61 dS/m and the TDS from 1625 to 1697 ppm. According to the classification proposed by the water quality criteria for emitter clogging [20], these values had a moderate potential for emitter clogging, as shown through the results of Capra and Scicolone, and Al-Mefleh and Al-Raja [12,23]. However, Al-Mefleh et al. indicated that the salt concentration in the TWW does not cause emitter clogging because the EC values of the TWW are low [21]. The mean values of Mn (0.50 mg/L), Ca (98.6 mg/L), and Fe (0.11 mg/L) have a low amount of emitter clogging potential.

The analysis of TWW shows that the mean value of the concentrations of $BOD_5$ (16 mg/L) has a low clogging potential of emitters, based on the classification by Capra and Scicolone, which may be attributed due to low concentration of suspended solids and organic matter [21,23]. Overall, the water quality analysis signifies the need for proper irrigation system design, operation, and maintenance when TWW is used and explains farmers' hesitation from utilizing TWW.

### 3.2. Emitter Characteristics

Initial values of emitter characteristics include the manufacturer emitter discharge (*Manuf.* $Q_{avg}$), initial emitter field discharge ($Q_{in}$), emitter discharge exponent ($X_{in}$), discharge coefficient ($Kd_{in}$), initial variation coefficient ($CV_{in}$), initial emission uniformity ($EU_{in}$), and Christiansen uniformity coefficient ($CU_{in}$). Table 3 displays these values for the

different emitter types. The initial average discharge of the emitters was measured in the field at varying operating pressure levels (0.25, 0.50, and 1.00 bar). Mostly, it was found that the $Q_{avg}$ increased in correlation with an increase in operational pressure. The final values of emitter characteristics consist of average discharge ($Q_{avg}$), emitter discharge exponent ($X_{fi}$), discharge coefficient ($Kd_{fi}$), coefficient of variation ($CV_{fi}$), emission uniformity ($EU_{fi}$), and Christiansen uniformity coefficient ($CU_{fi}$) at each treatment level. By comparing the initial and final values of $CV$ at the same operating pressure for each emitter, it was found that the final values of $CV$ decreased.

**Table 3.** The initial and final values of hydraulic parameters for emitter types LP and SP.

| Parameters | Emitter Type | | Pressure (bar) |
|---|---|---|---|
| | **LP** | **SP** | |
| Manuf. $Q_{avg}$ * (Lh$^{-1}$) | 8 | 8 | ** |
| Initial field $Q_{avg}$ (Lh$^{-1}$) | 7.0, 7.5, 7.9 | 7.0, 7.9, 7.6 | *** |
| $X_{in}$[a] | 0.082 | 0.062 | 0.25, 1 |
| $X_{fi}$[b] | 0.009 | 0.0012 | 0.25, 1 |
| $Kd_{in}$[c] | 8.359 | 7.262 | 0.25, 1 |
| $Kd_{fi}$ | 7.55 | 7.86 | 0.25, 1 |
| final field $Q_{avg}$ (Lh$^{-1}$) | 7.9, 7.6, 8.0 | 7.9, 7.9, 7.9 | *** |
| $CV_{in}$ | 0.3, 0.14, 0.11 | 0.34, 0.08, 0.17 | *** |
| $CV_{fi}$ | 0.1, 0.16, 0.08 | 0.11, 0.08, 0.08 | *** |
| $EU_{in}$ | 63, 83, 87 | 60, 89, 79 | *** |
| $EU_{fi}$ | 87, 82, 91 | 85, 90, 90 | *** |
| $CU_{in}$ | 79, 90, 92 | 77, 94, 88 | *** |
| $CU_{fi}$ | 93, 89, 94 | 91, 93, 94 | *** |

* Manufacturer average discharge of emitter; *** 0.25, 0.50, 1.00, respectively; ** 0.15 bar for LP and 1.00 bar for SP. [a] Initial exponent emitter discharge; [b] Final exponent emitter discharge; [c] Initial emitter discharge coefficient.

Additionally, in respect to the initial $EU$ values for each emitter at the same operational pressure, the final EU values increased. According to the classification by Özekici and Sneed, the $CV$ values were classified as excellent (<5%), average (5–7%), marginal (7–11%), poor (11–15%), and unacceptable (>15) [24]. This study found that the initial value of variation coefficient ($CV_{in}$) values were 0.3, 0.14, and 0.11 for the LP emitter, and 0.34, 0.08, and 0.17 for the SP emitter at operating pressures of 0.25, 0.50, and 1.00 bar, respectively. These results were considered unacceptable at 0.25 bar for both types of emitters, poor at 0.50 bar for the LP emitter, and marginal for the SP emitter. The results were marginal at 1.00 bar for the LP emitter and unacceptable for the SP emitter.

The final field emitter discharge of $Q_{avg}$, $X_{fi}$, $Kd_{fi}$, $CV_{fi}$, $EU_{fi}$, and $CU_{fi}$ were estimated after 160 h of operational time at the end of the experiment. The initial emitter discharge exponent ($X_i$) ranging from 0.00 to 1.00 tested the relation between the flow regime and operating pressure. Since the $X_i$ value is less than 0.5, less discharge is affected by changing the pressure, and vice versa if $X_i$ is greater than 0.5. Since the $X_i$ values for the proposed emitter are around 0.1, they are considered fully compensated emitters.

### 3.3. Average Emitter Discharge ($Q_{avg}$)

Two types of emitters, LP and SP, were used in order to measure the emitter discharge under three sets of operating pressures (P1 = 0.25, P2 = 0.50, and P3 = 1.00 bars) over a 160-h total operational time consisting of 20-h intervals using TWW. Statistical analysis was carried out for the average discharge ($Q_{avg}$) of each emitter type under different operating pressures and operational times. The main effect of emitter type on $Q_{avg}$ was not significant ($p < 0.05$). The main effect of interaction for the emitter type, operating pressure, and operational times on $Q_{avg}$ was significant ($p < 0.05$). The results of $Q_{avg}$ for the LP and SP emitters under different pressure values over the operational times are presented in Figure 2. For the LP emitter, the values of $Q_{avg}$ at 0.25 bar and operational times were

different from each other. However, the values of $Q_{avg}$ at each operating pressure of 0.50 bar and 1.00 bar over the operational times did not vary from one another. For the LP emitter at a pressure of 0.25 bar, the $Q_{avg}$ fell into three groups of operational times: the first group was 20 and 80 h, the second group was 60 and 160 h, and the third group was 40, 100, 120, and 140 h. The $Q_{avg}$ values were not different within each group, but they varied between the groups. For the LP emitter, the lowest values of $Q_{avg}$ were 7.03, 7.42, and 7.73 Lh$^{-1}$ at 0.25, 0.5, and 1.0 bar, respectively. While the maximum values of $Q_{avg}$ were 7.91, 7.83, and 8.00 Lh$^{-1}$, at 0.25, 0.50, and 1.00 bar, respectively.

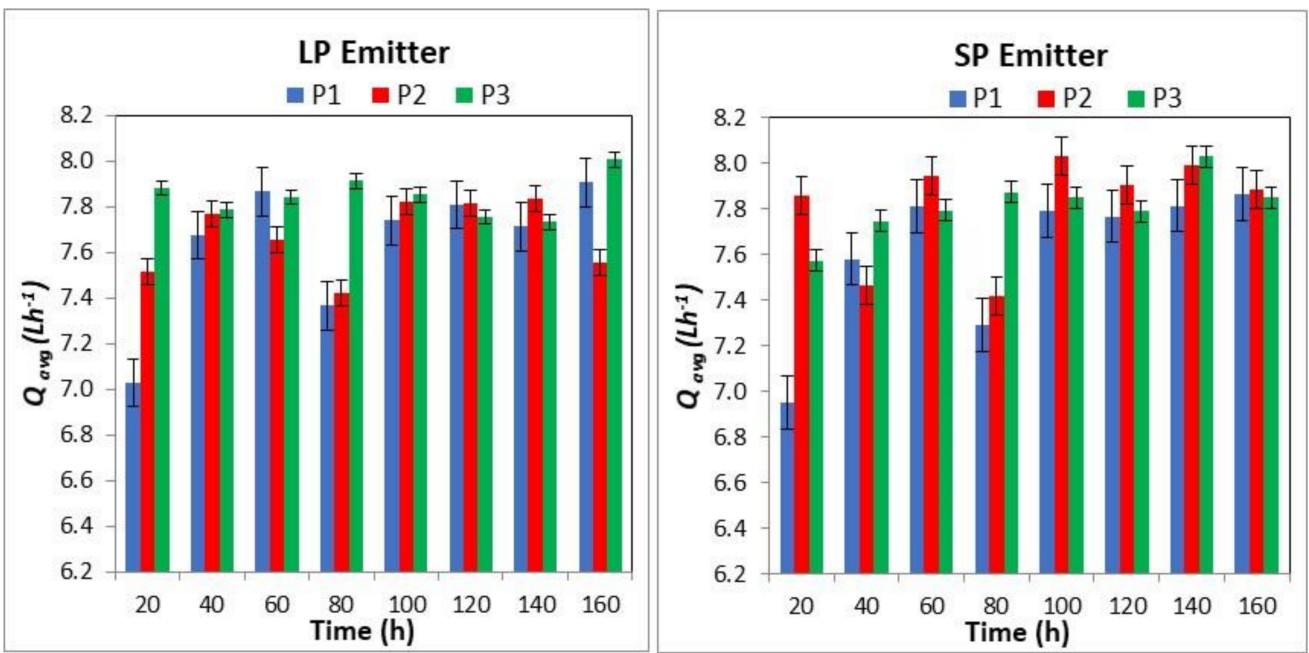

**Figure 2.** Average emitter discharge values ($Q_{avg}$) for LP and SP emitters using TWW under different operating pressures (P1 = 0.25, P2 = 0.50, and P3 = 1.00 bars) over varying operational times.

For the SP emitter, the behavior trend of $Q_{avg}$ values at each operational pressure and operational time was similar to $Q_{avg}$ values for the LP emitter. For the SP emitter at a pressure of 0.25 bar, the $Q_{avg}$ fell into the same three groups of operational times as the LP emitter. Similar to the LP emitter, the $Q_{avg}$ values were not different within each group but were different between the groups. For the SP emitter at 0.25 bar, the lowest values of $Q_{avg}$ were 6.95, 7.41, 7.57 Lh$^{-1}$ at 0.25, 0.50, and 1.0 bar, respectively. While the highest values of $Q_{avg}$ were 7.86, 8.00, 8.00 Lh$^{-1}$ at 0.25, 0.50, and 1.00 bar, respectively. This study noticed that emitter discharge increased slightly as the operating pressure is increased from 0.50 bar to 1.00 bar. The results of $Q_{avg}$ for both emitter types show that increasing the pressure can decrease the clogging potential of emitter discharge over the operational times.

For the LP emitter, the overall average emitter discharge (from eight measures with 20-h intervals) was 7.6, 7.7, and 7.8 Lh$^{-1}$ at 0.25, 0.50, and 1.00 bar, respectively. For the SP emitter, the overall average emitter discharge was 7.6, 7.8, and 7.8 Lh$^{-1}$ at 0.25, 0.50, and 1.00 bar, respectively. Based on these results, the deviations of emitter discharge for the LP emitter at 0.25, 0.50, and 1.00 bar from the recommended value (8 Lh$^{-1}$) by manufacturers were found to be 0.4, 0.3, and 0.2 Lh$^{-1}$, respectively. For the SP emitter at 0.25, 0.50, and 1.00 bar, the deviations from the recommended value (8 Lh$^{-1}$) were 0.4, 0.2, and 0.2 Lh$^{-1}$, respectively. With respect to manufacturing emitter discharge (8 Lh$^{-1}$) for the LP emitter, the $Q_{avg}$ (from eight measures with 20-h intervals) decreased by 5%, 3.75%, and 2.5% at 0.25, 0.50, and 1.0 bar, respectively. Additionally, for the SP emitter, the $Q_{avg}$ decreased by 5%, 2.5%, and 2.5% at 0.25, 0.50, and 1.0 bar, respectively. Based on the above results for both types of emitters (SP and LP), it was found that the $Q_{avg}$ (from eight measures

with 20-h intervals) at each operating pressure was reduced by 2% to 5% from the original values of emitter discharge tested by the manufacturer. For all of the emitter assessments, $Q_{avg}$ was above 7.0 Lh$^{-1}$ during the whole experiment, and they were 0.97 to 1.5 Lh$^{-1}$ lower than that of the manufacturing discharge (8.0 Lh$^{-1}$).

### 3.4. Coefficient of Variation (CV)

The *CV* values of the emitter types (LP and SP) under different operating pressure values (0.25, 0.50, and 1.00 bar) and over varying operational times are presented in Figure 3. The main effect of interaction for the emitter types, operating pressure, and operational time on the *CV* was significant ($p < 0.05$). Each combination of operating pressure with operational time and emitter type with operating pressure had a significant effect ($p < 0.05$) on the *CV*. However, the main effect of emitter type on *CV* values was not significant. Under each type of emitter (LP and SP), the *CV* values at an operating pressure of 1.00 bar and over different operational times were not different from each other. For the LP emitter at each operating pressure of 0.25 bar and 0.50 bar, *CV* values at 20 and 80 h were not different from each other, but they did vary from other values at different operational times. For the SP emitter at 0.25 bar, the *CV* values at 20 h of operational time were different from that of 80 h, and both of them were different from the *CV* values at the rest of the operational times. At 1.00 bar, the *CV* values at 40 h of operational time were different than that of 80 h, and both values were different from the *CV* values at the rest of the operational times. For the LP emitter, the *CV* values under 0.25, 0.50, and 1.00 bar over operational times varied from 10 to 30%, 13 to 16%, and 8 to 11%, respectively. The *CV* values of the SP emitter under 0.25, 0.50, and 1.00 bar over operational times varied from 11 to 33%, 8 to 22%, and 8 to 17%, respectively.

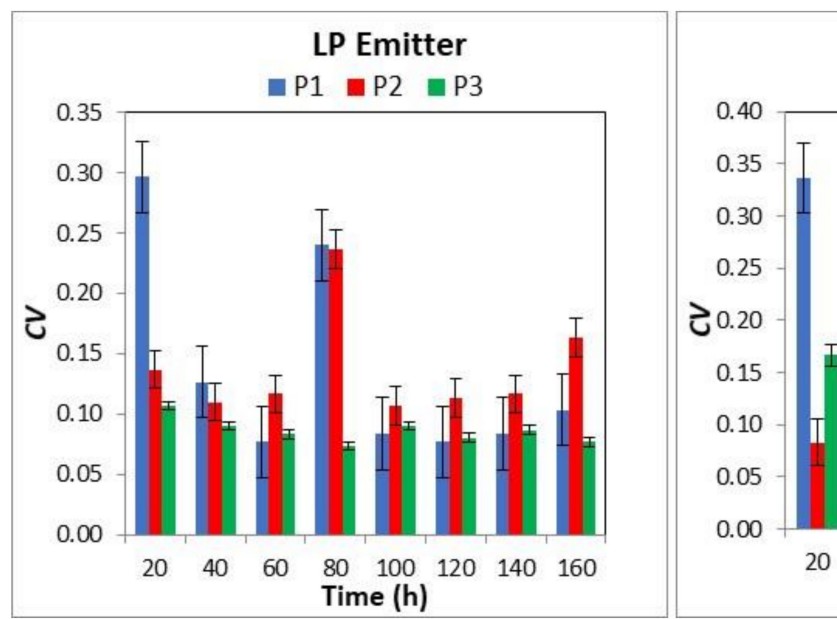 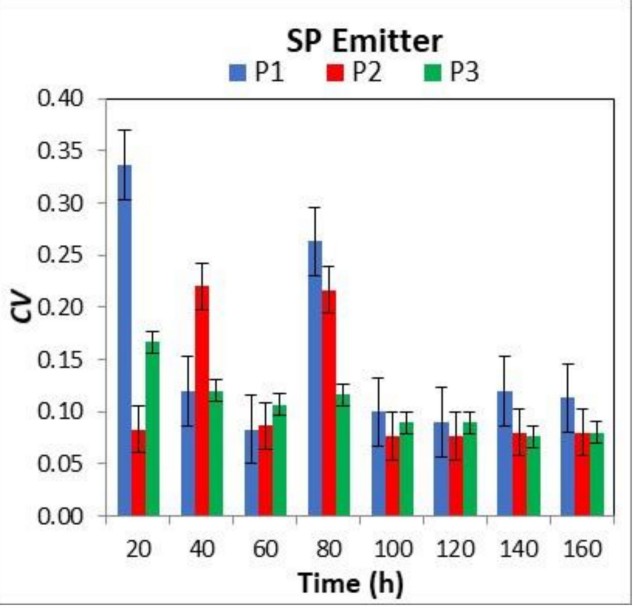

**Figure 3.** Coefficient of variation (*CV*) for LP and SP emitters using TWW under different operating pressures (P1 = 0.25, P2 = 0.50, and P3 = 1.00 bar) over different operational times.

Overall, the average values of *CV* in the eight tests for the LP emitter were 14, 14, and 9% at pressures of 0.25, 0.50, and 1.00 bar, respectively. The average values of *CV* for the SP emitter were 15, 12, and 11% at a pressure of 0.25, 0.50, and 1.00 bar, respectively. It was found that the *CV* values for LP and SP emitters at 0.25 bar are close to each other. According to Bralts and *American Society of Agricultural Engineers* (ASAE EP405.1), the values of *CV* for the LP emitter were classified as low, moderate, and good at pressures of 0.25, 0.50, and 1.00 bar, respectively. The good *CV* at the highest pressure might be attributed to

the fact that adequate pressure offsets the effects of factors affecting the variation in emitters discharge throughout the system, which is important to consider when designing irrigation systems, especially if TWW is used. The SP emitter *CV* values fell into the category of moderate at each pressure of 0.25, 0.50, and 1.00 bar, respectively [25,26]. This reflects a robust emitter manufacturing design and characteristics, which also indicates probably LP emitters' manufacturing characteristics need improvement in order to achieve consistent *CV* values at varying operational conditions.

　　　The time tests of *CV* results for LP and SP emitters under different operational pressures were classified according to Bralts and ASAE EP405.1 classification (Table 4) [25,26]. These classifications show that the majority of tests for the LP emitter are considered to be in the category of low to moderate, medium, and low at 0.25, 0.50, 1.00 bar, respectively. For the SP emitter, the majority of tests are considered to be in the category of moderate, low, and low to moderate at 0.25, 0.50, 1.00 bar, respectively. However, Hezarjaribi et al. stated that if the emitters' *CVs* are less than 5%, they give a realistic uniformity of water application [27]. Other studies indicated that the typical range values for *CVs* vary from 2 to 15% [28–30].

**Table 4.** Classification of the time tests of *CV* results for LP and SP emitters under different operational pressures according to Bralts and ASAE EP405.1 [26,27].

| Emitter Type | Pressure (bar) | Bralts | | | ASAE EP405.1 | | |
|---|---|---|---|---|---|---|---|
| | | L* | M* | H* | L* | M* | H* |
| LP | 0.25 | 5 | 2 | 1 | 4 | 2 | 2 |
| | 0.50 | 0 | 8 | 0 | 0 | 7 | 1 |
| | 1.00 | 7 | 1 | 0 | 7 | 1 | 0 |
| SP | 0.25 | 2 | 5 | 1 | 2 | 4 | 2 |
| | 0.50 | 6 | 2 | 0 | 6 | 0 | 2 |
| | 1.00 | 4 | 4 | 0 | 4 | 4 | 0 |

L* low, M* moderate, H* high.

### 3.5. Emission Uniformity (EU)

　　　The *EU* values of the LP and SP emitter types under three sets of operating pressures (0.25, 0.50, and 1.00 bar) and over varying operational times using TWW are presented in Figure 4. The interaction for the emitter type, different operating pressures, and operating time on *EU* was significant ($p > 0.05$). However, the main effect of emitter type and the combination of emitter type with operational times were not significant. For the LP emitter at 0.50 and 1.00 bar over different operational times (from 20 to 160 h, with an interval time of 20 h), the mean values of *EU* were not different from each other. Alternatively, for the LP emitter at 0.25 bar over the operational times, the mean values of *EU* varied from each other. This indicates that the performance of the LP emitter is stable at pressures equal and higher to 0.50 bar. Irrigation systems operating at a pressure less than 0.50 bar might be at risk of lower hydraulic performance unless measures are taken to reduce pressure loss throughout the system, especially toward the end of irrigation drip lines.

　　　The *EU* values over operational times of 20 and 80 h were different from each other, and both of them varied from other values at operational times of 40, 60, 100, 120, 140, and 160 h. For the LP emitter, the mean values of *EU* ranged from 63% to 91%, 74% to 87%, and 87% to 91% at 0.25, 0.50, and 1.00 bar, respectively. For the SP emitter at 0.25 bar, the *EU* values fell into two groups of operational times; the first group was 20 and 80 h, and the second group was 40, 60, 100, 120, 140, and 160 h. The mean values of *EU* were not different within the second group but were different in the first group. At a pressure of 0.50 bar, the *EU* value means fell into two groups of operational times; the first group was 40 and 80 h, and the second group was 20, 60, 100, 120, 140, and 160 h. The EU value means were not different within each group but varied between the groups. At 1.00 bar, the *EU* was not significantly different over the operational times. For the SP emitter, the values of *EU* varied from 60% to 90%, 75% to 92%, and 79% to 91% at 0.25, 0.50, and 1.00 bar,

respectively. The average values of *EU* over eight tests for the LP emitter were 84%, 84%, and 89% at pressures of 0.25, 0.50, and 1.00 bar, respectively. The average values of *EU* for the SP emitter were 82%, 82%, and 87% at pressures of 0.25, 0.50, and 1.00 bar, respectively. These results show that the *EU* values for LP emitters are higher by 2% than those values of SP emitters at each operational pressure (0.25, 0.50, and 1.00 bar). This highlights the advantage of LP emitters over SP emitters. Low activation pressure requirement is more likely to result in a higher irrigation distribution uniformity. However, other factors such as clogging potential must be considered too.

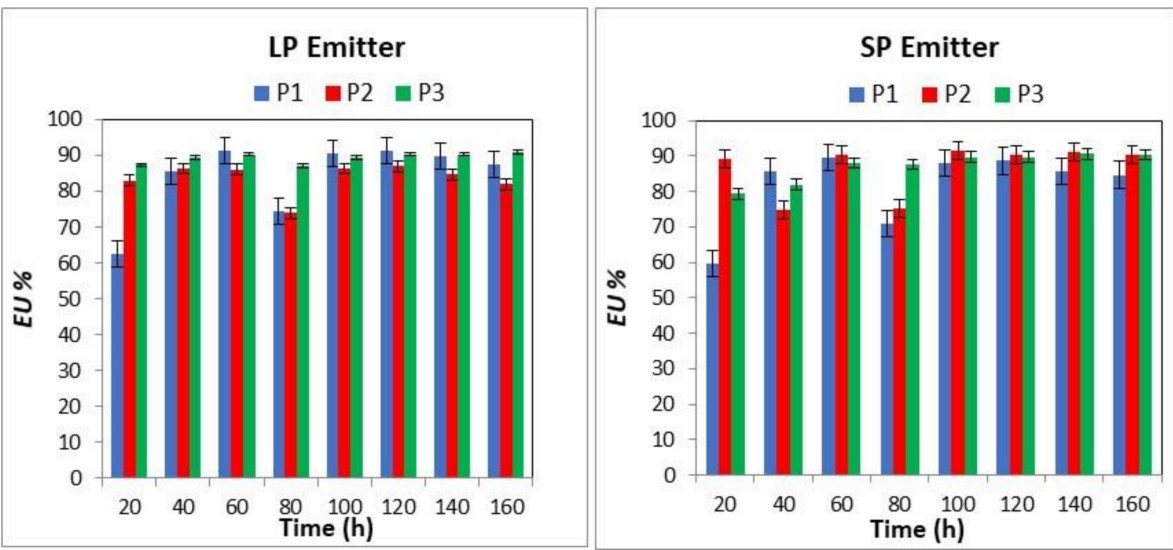

**Figure 4.** The emission uniformity (*EU*) values for LP and SP emitters using TWW under different operating pressures (P1 = 0.25, P2 = 0.50, and P3 = 1.00 bars) over varying operational times.

According to the classification by Keller and Bliesner and ASAE EP405.1 (Table 5), the majority of the operational time tests (eight tests) performed of the *EU* results for LP and SP emitters under varying operational pressures fall into the category of high performance [26,31]. Referring to the classification by Keller and Bliesner, the results of this study indicate that the *EU* values at 0.25 bar for the LP and SP emitters varied from low to moderate, while the *EU* values at 0.50 and 1.00 bar were high [31], which indicated that there is still a need to improve the design and operational characteristics of LP emitters. Based on the classification by ASAE EP405.1, the results of this study indicate that the *EU* values for the LP and SP emitters at an operational pressure of 1.00 bar are more suitable for irrigation with TWW [26]. There have been several studies [21,22,27] that have dealt with the effect of TWW on emitter performance and found that increasing EU values leads to decreasing *CV* values, which is consistent with our results.

**Table 5.** Classification of time tests of the *EU* results for LP and SP emitters under different operational pressures according to Keller and Bliesner and ASAE EP405.1 [26,31].

| Emitter Type | Pressure (bar) | Keller and Bliesner | | | ASAE EP405.1 | | |
|---|---|---|---|---|---|---|---|
| | | L* | M* | H* | L* | M* | H* |
| LP | 0.25 | 1 | 1 | 6 | 0 | 2 | 6 |
| | 0.50 | 0 | 1 | 7 | 0 | 1 | 7 |
| | 1.00 | 0 | 0 | 8 | 0 | 0 | 8 |
| SP | 0.25 | 0 | 2 | 6 | 0 | 2 | 6 |
| | 0.50 | 0 | 2 | 6 | 0 | 2 | 6 |
| | 1.00 | 0 | 0 | 8 | 0 | 1 | 7 |

L*: low. M*: medium. H*: high.

### 3.6. Christiansen Uniformity (CU)

The Christiansen uniformity (*CU*) values for the LP and SP emitters under three sets of operating pressures (0.25, 0.50, and 1.00 bar) and over an operational time interval of 20 h using TWW are presented in Figure 5. The main effect of the interaction of the emitter type, different operating pressures, and operational times on *CU* was significant. The effect of the emitter type on *CU* was not significant. The means of *CU* values were significantly different under operating pressures and operational times. For the LP emitter at 0.25 bar at different operational times, the mean values of *CU* were different from each other. However, the mean values of the LP emitter at 0.5 and 1.00 bar over varying operational times were not different from each other. For the LP emitter, the *CU* values varied from 79 to 95%, 85 to 92%, and 92 to 94% at operating pressures of 0.25, 0.50, and 1.00 bar, respectively. At 0.25 bar, the *CU* values fell into two groups of operational times; the first group was 20 and 80 h, and the second group was 40, 60, 100, 120, 140, and 160 h. The *CU* values were not different in the second group but were different in the first group.

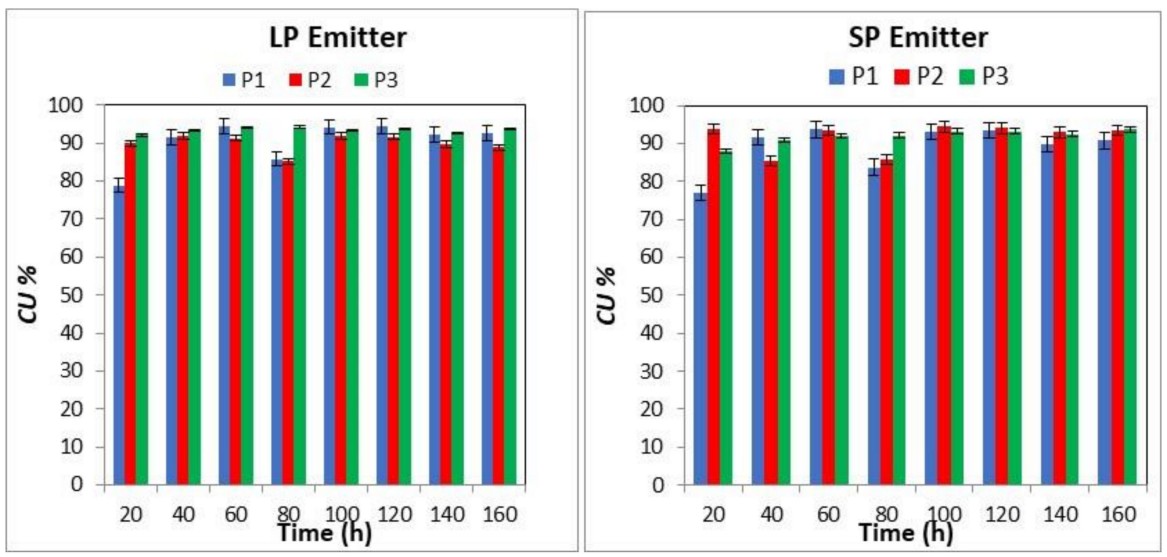

**Figure 5.** Christiansen uniformity (*CU*) values for LP and SP emitters using TWW under different operating pressures (P1 = 0.25, P2 = 0.50, and P3 = 1.00 bars) over different operational times.

For the SP emitter, the mean values of *CU* at operating pressures of 0.25 and 0.50 bar and an operational time of 20-h intervals were different from each other. However, the mean values of *CU* at an operating pressure of 1.00 bar over varying operational times were not different from each other. At 0.25 bar, the *CU* values demonstrated the same results as the LP emitter, as mentioned above. At a pressure of 0.50 bar, the *CU* values fell into two groups of operational times; the first group was 40 and 80 h, and the second group was 20, 60, 100, 120, 140, and 160 h. The *CU* values were not different in each group but were different between the groups. For the SP emitter, *CU* values varied from 78 to 94%, 85 to 94%, and 88 to 93% at operating pressures of 0.25, 0.50, and 1.00 bar, respectively. According to the Keller and Bliesner classification, the results of this study indicate that the *CU* under 0.25 bar for the LP and SP emitter types that fell in the first group of operational time (20 and 80 h intervals) varied from low to moderate, respectively, and the second group (40, 60, 100, 120, 140, and 160 h) was high [31]. Overall, the results showed that with increased pressure, *CU* values would increase over the operational time.

The average values of *CU* over eight tests for the LP emitter were 91%, 90%, and 93% at pressures of 0.25, 0.50, and 1.00 bar, respectively. The average values of *CU* for the SP emitter were 89%, 92%, and 92% at pressures of 0.25, 0.50, and 1.00 bar, respectively. It was found that the difference in the *CU* values varied from 1 to 2% at each operational pressure for LP and SP emitters, respectively. According to the classification by Keller and Bliesner and

ASAE EP405.1 (Table 6), the majority of the time tests of *CU* values for LP and SP emitters under different operational pressures were classified as high-performance emitters [26,31]. *CU* results, when compared with the *EU* result, indicate that the *EU* indicator is more suitable for evaluation and assessment of the performance of LP emitters than the *CU*. Additionally, this derives from the core difference between the two approaches, i.e., the *EU* approach focuses on the lower quarter of emitter discharges, while the *CU* approach looks at the average discharge of all emitters. Therefore, we recommend practitioners concentrate on the use of *EU* when assessing LP emitters.

**Table 6.** Classification of the time tests of *CU* results for LP and SP emitters under different operational pressures according to Keller and Bliesner and ASAE EP405.1 [26,31].

| Emitter Type | Pressure (bar) | Keller and Bliesner | | | ASAE EP405.1 | | |
|---|---|---|---|---|---|---|---|
| | | L* | M* | H* | L* | M* | H* |
| LP | 0.25 | 1 | 1 | 6 | 0 | 2 | 6 |
| | 0.50 | 0 | 1 | 7 | 0 | 1 | 7 |
| | 1.00 | 0 | 0 | 8 | 0 | 0 | 8 |
| SP | 0.25 | 0 | 2 | 6 | 0 | 2 | 6 |
| | 0.50 | 0 | 2 | 6 | 0 | 2 | 6 |
| | 1.00 | 0 | 0 | 8 | 0 | 1 | 7 |

L*: low. M*: medium. H*: high.

### 3.7. The Main Degree of Clogging (DC)

The main degree of clogging (*DC*) values of the emitter types (LP and SP) under three sets of operating pressures (0.25, 0.50, and 1.00 bar) over an operational time interval of 20 h was estimated using TWW, which are presented in Figure 6. The interaction of emitter types under different operating pressures (0.25, 0.50, and 1.00 bar) and operational times on *DC* was not significant ($p < 0.05$). For the LP emitter, the mean values of *DC* at each operating pressure (0.25, 0.50, and 1.00 bar) and each operational time were not different from each other; the *DC* values varied from 0.0 to 11%, 0.0 to 4%, and 0.0 to 1.5%, at operating pressures of 0.25, 0.50, and 1.00 bar, respectively. The average values of *DC* over eight tests for the LP emitter were 8%, 2%, and 1% at pressures of 0.25, 0.50, and 1.00 bar, respectively.

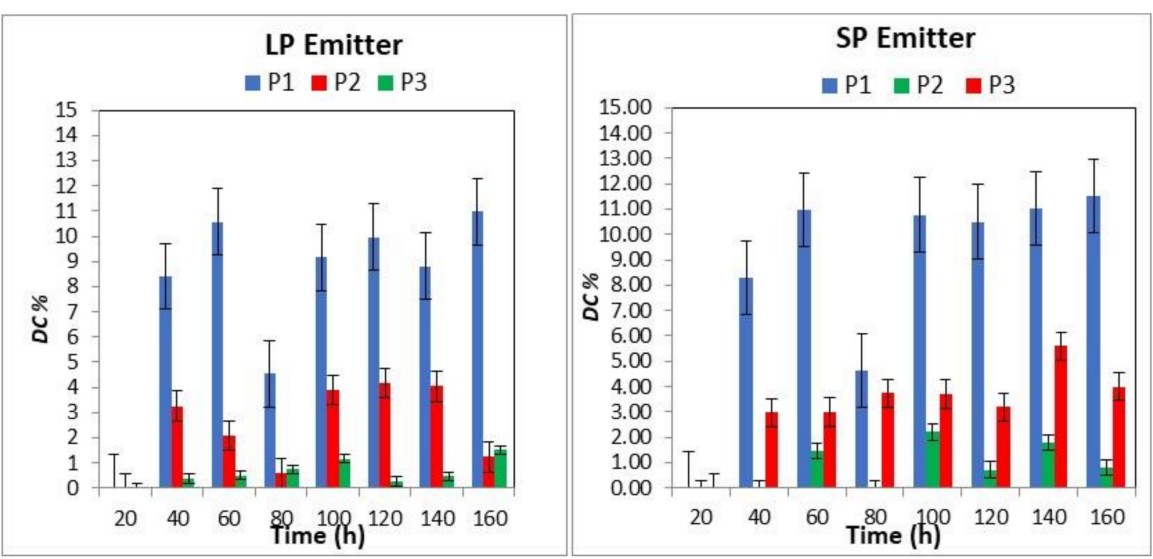

**Figure 6.** The mean degree of clogging (*DC*) for LP and SP emitters using TWW under different operating pressures (P1 = 0.25, P2 = 0.50, and P3 = 1.00 bars over different operational times.

For the SP emitter, the mean values of *DC* at each operating pressure (0.25 and 1.00 bar) and each operational time were different from each other, while they were not different at an operating pressure of 0.50 bar. At 0.25 bar, the *DC* values fell into two groups of operational times—the first group being 40 and 80 h and the second group being 60, 100, 120, 140, and 160 h. The mean values of *DC* were not different within each group but were different between the groups. The *DC* values varied from 0.0 to 12%, 0.0 to 2%, and 0.0 to 6% at operating pressures of 0.25, 0.50, and 1.00 bar, respectively. The average values of *DC* for the SP emitter over eight tests were 8.5%, 0.9%, and 3.1% at pressures of 0.25, 0.50, and 1.00 bar, respectively. The impact of pressure on the *DC* for both emitter types is similar, and this shows that as the operating pressure increases, the *DC* drops. The increasing pressure might be responsible for creating an environment in the emitter in which the deposition of sediment particles is less likely to occur. This hints at the importance of incorporating an adequate filtration system, particularly when operating at low pressures.

## 4. Conclusions

The results of this study showed that the main effect of the LP and SP emitter types on emitter discharge was not significant. However, the operating pressure and operating time had a significant effect on the $Q_{avg}$, *CV*, *EU*, *CU*, and *DC*.

The deviations of emitter discharge for the LP emitter at 0.25, 0.50, and 1.00 bar from the recommended value (8 Lh$^{-1}$) by manufacturers of the emitters were found to be 0.4, 0.3, and 0.2 Lh$^{-1}$, respectively. For the SP emitter at 0.25, 0.50, and 1.00 bar, the deviations from the recommended value (8 Lh$^{-1}$) were 0.4, 0.2, and 0.2 Lh$^{-1}$, respectively. For all of the emitter assessments, $Q_{avg}$ was above 7.0 Lh$^{-1}$ during the whole experiment, and it was appropriately 0.97 to 1.5 Lh$^{-1}$ lower than that of the manufacturing discharge (8.0 Lh$^{-1}$). The *CV* values show that the majority of tests for the LP emitter are considered to be in the categories of low to moderate, medium, and low at 0.25, 0.50, and 1.00 bar, respectively. The *CV* values for the SP emitter are considered to be in the categories of moderate, low, and low to moderate at 0.25, 0.50, and 1.00 bar, respectively.

The results of this study indicate that the *EU* values at 0.25 bar for LP and SP emitters varied from low to moderate, while the *EU* values at 0.50 and 1.00 bar were high. Overall, it was found that the *EU* values at 1.00 bar for LP and SP emitters are more recommended. Additionally, the *EU* values indicate that LP and SP emitters are more suitable to be used with TWW at an operational pressure of 1.00. The main degree of clogging did not increase significantly when the operating pressure increased from 0.25 to 1.00 bar. However, it increased dramatically when the operating pressure decreased to 0.25 bar. Overall, we recommend using the tested emitters (LP and SP) under TWW since there was no significant difference in the emitter discharge, and their deviations from the recommended discharge (8 Lh$^{-1}$) varied from 0.4 to 0.2 Lh$^{-1}$ as the pressure changed from 0.25 to 1.00 bar.

In conclusion, LP emitters are a promising addition to the irrigation sector because they aid in the overall reduction of energy requirements of irrigation systems. However, utilization of LP emitters, as shown by this study, calls for greater attention to the filtration system in place and favors the use of the *EU* approach over the *CU* approach when monitoring the irrigation distribution uniformity.

**Author Contributions:** Conceptualization, N.K.A.-M., S.T.; methodology, N.K.A.-M., S.T.; formal analysis, N.K.A.-M., S.T.; investigation, N.K.A.-M., S.T., K.A.N.; resources, N.K.A.-M., S.T.; data curation, N.K.A.-M., S.T., K.A.N.; writing—original draft preparation N.K.A.-M., S.T., K.A.N.; writing—review and editing, N.K.A.-M., S.T. All authors have read and agreed to the published version of the manuscript.

**Funding:** This research was supported by Jordan University of Science and Technology and National Center for Agriculture Research and Extension, Jordan.

**Institutional Review Board Statement:** Not applicable.

**Informed Consent Statement:** Not applicable.

**Data Availability Statement:** Not applicable.

**Acknowledgments:** The authors would like to express their gratitude and appreciation to the Jordan University of Science and Technology for their funding and supporting this research. Appreciations are extended to the National Center for Agriculture Research and Extension for their assistance and support to conduct this research in Al-Ramtha Agricultural Research Station.

**Conflicts of Interest:** The authors declare no conflict of interest.

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
