# Peer review of "Assessment of Treated Wastewater Reuse in Drip Irrigation under Different Pressure Conditions"

_water, doi:10.3390/w13081033_

Round 1
Reviewer 1 Report
Dear Authors,
Attached please find some comments regarding the manuscript. First of all, please verify the discussion of the results (it would be optimal to separate it from the results).
The number of references over the age of 10 is absolutely unacceptable. The discussion of the results must be supported by the latest literature (not only your authorship).
Just because of the apparent effort put into the analysis, the manuscript was not rejected by me at this stage.
Yours faithfully,
Rev

Author Response
Dear reviewer
Thank you for your efforts. The comments and responses are presented in the following table.

Reviewer 2 Report
Dear authors, the selected topic is surely relevant. And the chosen method is generally able to answer the research question. However, the paper suffers badly from a purely descriptive and exhausting listing of observations. The conclusion (both emitter types are applicable) is really banal.
Right in the beginning, I miss already clearly derived research question which lead us through the methods, results and conclusions. If the focus is on the comparison of different emitter types one should at least explain the differences between both and derive from them the research question.
And you could have derived so much more out of the acquired data, asking the Why-question. I miss completely any process oriented discussion. What are the potential reasons for your findings, concerning effect of operation time, pressure and emitter type. Instead you list in an uninspired way dozens of numbers for the partly redundant criteria. E.g., from a physical point of view, I could find several potential reasons, why higher pressure lead to more uniformity and lower clogging risk. Why are in some repeating experiments the flows lower than in others? not reproducible cleaning ???
I suggest, that you revise the result and discussion chapter completely. Replace the pure listing of numbers in the text by meaningful diagrams in combination with a process oriented discussion. Then is can become a meaningful paper.
Check also the comments I made in the pdf. Since, your way of discussion repeats again and again for each criterion I did not comment each. But my comments apply in same tenor to all criteria.

Author Response

(The authors gave the same response as above.)

Round 2
Reviewer 1 Report
Dear Authors,
thank you for the corrections made.
Yours faithfully,
Author Response
Dear Reviewer,
Thank you for all your comments.
Reviewer 2 Report
Thank you for improving readability and understanding of the manuscript.
However, the discussion still suffers from a pure empiric observation without the interest of understanding the causal relationships. So far, you present the different assessment criteria. Accordingly, the drawn conclusion are correct but rather banal.
I answered on your answers at some points. This might help you to better understand what I mean. I really would like you to perform a more meaningful discussion about the potential reasons of your findings. This would sinificantly improve the attrraction to potential readers.

Author Response

(The authors gave the same response as above.)
